# Gypenoside A from *Gynostemma pentaphyllum* Attenuates Airway Inflammation and Th2 Cell Activities in a Murine Asthma Model

**DOI:** 10.3390/ijms23147699

**Published:** 2022-07-12

**Authors:** Wen-Chung Huang, Shu-Ju Wu, Kuo-Wei Yeh, Chian-Jiun Liou

**Affiliations:** 1Graduate Institute of Health Industry Technology, Research Center for Food and Cosmetic Safety, Chang Gung University of Science and Technology, Taoyuan City 33303, Taiwan; wchuang@mail.cgust.edu.tw; 2Division of Allergy, Asthma, and Rheumatology, Department of Pediatrics, Chang Gung Memorial Hospital, Linkou, Guishan Dist., Taoyuan City 33303, Taiwan; kwyeh@cgmh.org.tw; 3Department of Pediatrics, New Taipei Municipal TuCheng Hospital (Built and Operated by Chang Gung Medical Foundation), New Taipei 23656, Taiwan; 4Department of Nutrition and Health Sciences, Research Center for Chinese Herbal Medicine, Chang Gung University of Science and Technology, Taoyuan City 33303, Taiwan; sjwu@mail.cgust.edu.tw; 5Aesthetic Medical Center, Department of Dermatology, Chang Gung Memorial Hospital, Linkou, Taoyuan City 33303, Taiwan; 6Department of Nursing, Division of Basic Medical Sciences, Research Center for Chinese Herbal Medicine, Chang Gung University of Science and Technology, Taoyuan City 33303, Taiwan

**Keywords:** airway hyperresponsiveness, asthma, gypenoside A, T helper cells, tracheal epithelial cells

## Abstract

Our previous study found that oral administration of *Gynostemma pentaphyllum* extract can attenuate airway hyperresponsiveness (AHR) and reduce eosinophil infiltration in the lungs of asthmatic mice. Gypenoside A is isolated from *G. pentaphyllum*. In this study, we investigated whether gypenoside A can effectively reduce asthma in mice. Asthma was induced in BALB/c mice by ovalbumin injection. Asthmatic mice were treated with gypenoside A via intraperitoneal injection to assess airway inflammation, AHR, and immunomodulatory effects. In vitro, gypenoside A reduced inflammatory and oxidative responses in inflammatory tracheal epithelial cells. Experimental results showed that gypenoside A treatment can suppress eosinophil infiltration in the lungs, reduce tracheal goblet cell hyperplasia, and attenuate AHR. Gypenoside A significantly reduced Th2 cytokine expression and also inhibited the expression of inflammatory genes and proteins in the lung and bronchoalveolar lavage fluid. In addition, gypenoside A also significantly inhibited the secretion of inflammatory cytokines and chemokines and reduced oxidative expression in inflammatory tracheal epithelial cells. The experimental results suggested that gypenoside A is a natural compound that can effectively reduce airway inflammation and AHR in asthma, mainly by reducing Th2 cell activation.

## 1. Introduction

Asthma is a chronic allergic airway disease characterized by peribronchial inflammation, airway hyperresponsiveness (AHR), and airway remodeling [1]. During asthma attacks, airway smooth muscles contract and tracheal goblet cells secrete excess mucus, which obstructs the airway [2]. As a result, patients experience shortness of breath, wheezing, and dry cough. Persistent dyspnea will lead to suffocation and death [1]. Current clinical asthma treatment and preventive drugs mainly include glucocorticoids, bronchodilators, and long-acting β2-adrenergic receptor agonists [3]. However, the incidence of asthma remains high and is increasing worldwide [4]. Recent studies have shown that long-term inhaled or oral glucocorticoids also reduce the immune response against various pathogens [5]. Therefore, a novel strategy is needed to develop new therapeutic methods and drugs based on the pathogenesis of asthma.

Asthma attacks and worsening of pathological symptoms are closely related to immune system imbalance [6]. Activated Th2 cells release large amounts of cytokines including IL-4, IL-5, and IL-13 [7]. These cytokines stimulate the infiltration of activated eosinophils and mast cells in the lungs and the release of inflammatory and oxidative molecules from these immune cells, aggravating the inflammatory response in the lungs and increasing AHR, smooth muscle contraction, and tracheal goblet cell hyperplasia for mucus hypersecretion [6]. Blocking excessive Th2 cell activation in the respiratory system is thus a novel strategy for improving asthma and attenuating airway inflammation and allergic reactions.

*Gynostemma pentaphyllum* is a perennial plant of the Cucurbitaceae family, mainly grown in southern China, Taiwan, Japan, and Korea [8]. Traditional Chinese medicine uses *G. pentaphyllum* to treat hypertension and hyperlipidemia [9]. Recent studies have found that *G. pentaphyllum* has multiple pharmacological effects, including the treatment of hepatitis, stomach ulcers, and cancer [10,11]. *G. pentaphyllum* can also regulate hyperlipidemia and blood sugar [12]. Our previous experiments found that *G. pentaphyllum* extract could improve asthma in asthmatic mice [13,14]. However, it is not clear which active compounds of *G. pentaphyllum* might improve airway inflammation or oxidative stress in asthmatic mice. This study was undertaken to evaluate the ability of gypenoside A, a triterpenoid isolated from *G. pentaphyllum* [15], to suppress the activation of Th2 cells and thereby mitigate airway inflammation, AHR, and excessive mucus secretion in asthmatic mice.

## 2. Results

### 2.1. Gypenoside A Attenuated Eosinophil Infiltration and Goblet Cell Hyperplasia 

Compared with the OVA group mice, asthmatic mice treated with gypenoside A or prednisolone had reduced eosinophil infiltration of the lungs (Figure 1A), and gypenoside A-treated mice had lower inflammatory pathology scores (Figure 1B). PAS staining demonstrated that, compared with the OVA group, asthmatic mice treated with gypenoside A had less tracheal goblet cell hyperplasia (Figure 1C,D). Moreover, *Muc5Ac* gene expression in the lung was suppressed in the gypenoside A-treated asthmatic mice relative to the level seen in the OVA group mice (Figure 1E). 

### 2.2. Gypenoside A Mitigated AHR and Eosinophil Infiltration in Bronchoalveolar Lavage Fluid

At 40 mg/mL methacholine, asthmatic mice treated with gypenoside A or prednisolone had significantly reduced Penh values when compared with the OVA group mice (Figure 2A). Mice were anesthetized and bronchoalveolar lavage fluid (BALF) was collected as described previously [16]. Compared with the OVA group mice, asthmatic mice treated with prednisolone or gypenoside A also had significantly decreased numbers of eosinophils in BALF (Figure 2B). Treatment with gypenoside A or prednisolone also decreased the total number of cells in BALF (Figure 2B).

### 2.3. Gypenoside A Regulates Chemokine and Cytokine Secretion in BALF

Compared with the OVA group, the GPA10 and GPA30 groups had significantly reduced IL-4, IL-5, IL-13, TNF-α, IL-6, CCL11, and CCL24 expression (Figure 3A–G); conversely, the GPA30 group had elevated interferon (IFN)-γ levels (Figure 3H).

### 2.4. Gypenoside A Regulates Serum Antibodies and Splenocyte Cytokine Levels

Gypenoside A effectively reduced the levels of OVA-IgE and OVA-IgG_1_, and raised OVA-IgG_2a_ levels in the serum of asthmatic mice (Figure 4A–C). Gypenoside A remarkably reduced IL-4, IL-5, and IL-13 levels in the supernatant of splenocytes, and raised IFN-γ secretion relative to that in the OVA group mice (Figure 4D–G). Furthermore, Gypenoside A also did not show increased ALT and AST levels in asthmatic mice (Figure 4H–I).

### 2.5. Gypenoside A Modulates Antioxidant Enzyme and Inflammatory Gene Expression in Lung

Gypenoside A promoted glutathione (GSH) and suppressed malondialdehyde (MDA) activity in the lungs of asthmatic mice (Figure 5A,B). Gypenoside A also clearly reduced *TNF*, *IL6*, and *COX2* expression in the lungs of asthmatic mice (Figure 5C–E).

### 2.6. Gypenoside A Mitigates Inflammation and the ROS Response in BEAS-2B Cells

The cytotoxicity of gypenoside A in BEAS-2B cells was determined using the CCK8 assay. Gypenoside A did not demonstrate significant cytotoxic effects at a concentration ≤20 μM, and subsequent experiments used gypenoside A at 0–10 μM (Figure 1A). Gypenoside A effectively reduced IL-6, IL-8, MCP-1, CCL5, CCL11, and CCL24 secretion in TNF-α /IL-4–activated BEAS-2B cells (Figure 6B–G). Fluorescence microscopy showed that gypenoside A--treated BEAS-2B cells had a clearly attenuated ROS response compared with that of activated BEAS-2B cells (Figure 7A,B). Next, the DCFH-DA assay results demonstrated that gypenoside A suppressed ROS levels in IL-4/TNF-α stimulated BEAS-2B cells (Figure 7C).

## 3. Discussion

Our previous study found that long-term or short-term oral administration of *G. pentaphyllum* extract could attenuate airway inflammation and eosinophil infiltration in the lungs of OVA-sensitized mice by reducing Th2 cell activity [13,14]. Others have found that gypenosides III and VIII isolated from *G. pentaphyllum* reduce the bronchoconstrictor response in histamine-sensitized guinea-pigs [17]. In the past few years, more gypenosides have been isolated from *G. pentaphyllum* [8,18]; however, it is unclear whether these gypenosides can relieve asthma. In this study, we found that gypenoside A can inhibit eosinophil infiltration in the lung, weaken AHR, and reduce the expression of Th2-related cytokines and chemokines in BALF of asthmatic mice. Gypenoside A also suppressed tracheal goblet cell hyperplasia and attenuated *Muc5AC* expression, suppressing the mucus hypersecretion that can cause asphyxia. In addition, gypenoside A also significantly reduced inflammatory cytokine and chemokine expression in activated tracheal epithelial cells.

In patients with asthma, activated Th2 cells can release more cytokines to induce more allergic and inflammatory cell infiltration into the airway and lung tissue, causing AHR and clinical symptoms of asthma [6]. IL-13 knockout mice induced asthma, and the mice did not increase AHR and goblet cell hyperplasia [19]. Airway treatment of asthmatic mice with an anti- IL-13 antibody can reduce AHR, airway inflammation, and airway remodeling [19]. Gypenoside A inhibited AHR in asthmatic mice, and IL-13 levels were significantly reduced in BALF and spleen cell culture medium. We therefore think that gypenoside A inhibited Th2 cell activation and IL-13 secretion, thereby blocking AHR in OVA-sensitized mice.

Activated Th2 cells release excess IL-5, which can increase the differentiation of bone marrow cells to form more eosinophils [6]. Inflamed airway epithelial cells release large amounts of eotaxins (CCL11 and CCL24) that attract these eosinophils to migrate and infiltrate lung tissue [20]. In asthmatic mice, anti-IL-5 antibody treatment can reduce AHR and eosinophil infiltration in lung tissue [7]. In our experiments, gypenoside A reduced IL-5 levels in BALF and spleen cell culture medium. Gypenoside A treatment also suppressed CCL11 and CCL24 production by tracheal epithelial cells. Thus, gypenoside A can suppress eosinophil infiltration in the lungs of asthmatic mice. Activated eosinophils release eosinophil cationic protein and eosinophil peroxidase, leading to lung damage and inflammation [7]. Activated macrophages also release more inflammatory mediators and cytokines, increasing the severity of lung inflammation in asthmatic mice [6]. Gypenoside A treatment of TNF-α /IL-4-activated BEAS-2B cells reduced their expression of chemokines and inflammatory cytokines, suggesting that it could inhibit local airway inflammation in asthmatic mice. Gypenoside A also decreased *IL6*, *TNF-α*, and *COX2* gene expression in the lungs. Therefore, gypenoside A can reduce airway inflammation by inhibiting the expression of inflammatory cytokines and chemokines and pulmonary eosinophil infiltration in asthmatic mice.

In patients with asthma, tracheal goblet cells undergo hyperplasia and secrete excess mucus that obstructs the airway [20]. Mucins are glycoproteins produced by airway epithelial cells to clear allergens or microorganisms that invade the respiratory system [21]. In double IL-4/IL-13 knockout mice, the induction of asthma did not promote tracheal goblet cell hyperplasia and mucin secretion [22]. Our results confirm that gypenoside A reduces goblet cell hyperplasia by blocking the production of IL-13 and IL-4 in OVA-sensitized asthmatic mice, thereby reducing *Muc5AC* expression and mucus hypersecretion. In addition, IL-4 can also stimulate IgE secretion from B cells [6]. Allergen-IgE binding to mast cells stimulates mast cell activation and the consequent release of large amounts of histamine and leukotrienes, leading to severe allergic and inflammatory responses in the respiratory system [23]. We think that gypenoside A may inhibit the production of IgE in serum by blocking IL-4 expression in asthmatic mice.

Inflammatory immune cells in the lungs of patients with asthma also release more oxidative stress molecules and stimulate airway epithelial cells to release more inflammatory cytokines and ROS [23,24]. Activated eosinophils release eosinophil peroxidase, which causes oxidative damage in patients with asthma [25]. Oxidative stress can increase tracheal constriction, stimulate mucus secretion, and increase shortness of breath and dyspnea [24]. ROS can also cause apoptosis and DNA damage in alveolar cells, worsening lung function in patients with asthma [26]. In our experiments, gypenoside A significantly increased GSH levels and decreased MDA levels in the lungs of asthmatic mice. Our experiments confirm that gypenoside A is an effective antioxidant that can reduce allergen-induced lung cell damage in asthmatic mice.

## 4. Materials and Methods

### 4.1. Materials

We prepared gypenoside A (purity ≥ 98%, ChemFaces, Wuhan, China) in a stock solution of 30 mM in DMSO. In experiments using cultured cells, the DMSO concentration was less than 1% in the culture medium. In animal experiments, gypenoside A was dissolved in DMSO, and doses of 10 mg/kg and 30 mg/kg gypenoside A were prepared in a final volume of 50 µL.

### 4.2. Animals

The 6-week-old female BALB/c mice were purchased from the National Laboratory Animal Center (Taipei, Taiwan). Initially, these mice were housed under standard laboratory conditions for 7 days to acclimate to the environment of the animal house, and provided with water and standard chow ad libitum. The Institutional Animal Care and Use Committee of Chang Gung University of Science and Technology approved all the experimental animal protocols (IACUC approval number: 2020-001) and NIH Guides for the Care and Use of Laboratory Animals.

### 4.3. Mouse Experimental Procedure

Mice were randomly divided into 5 groups: a normal saline control group (N group) (n =8), an ovalbumin (OVA)-induced asthma mice group (OVA group) (n = 8); a 5 mg/kg prednisolone group (P group) (n = 8); and gypenoside-A 10 mg/kg and 30 mg/kg groups (the GPA10 and GPA30 groups, respectively) (n = 8 in each group). Mice were sensitized on days 1–3 and 14 by intraperitoneal injection of sensitization solution (0.8 mg AlOH3 and 50 μg OVA in 200 μL PBS). Mice were challenged on days 14, 17, 21, 24, and 28 with inhaled atomized OVA solution from an ultrasonic nebulizer. Prednisolone or gypenoside A was administered 1 h before the OVA challenge or methacholine inhalation (day 28), by intraperitoneal injection. On day 29, the end of the animal experiment, the mice were anesthetized and sacrificed, and tissues were removed for experimental analysis of asthma pathology, inflammation, and immune regulation, as previously described [27].

### 4.4. Measurement of Airway Hyperresponsiveness

To evaluate the pulmonary function of asthmatic mice, whole-body plethysmography (Buxco Electronics, Troy, NY, USA) was used to detect AHR in mice. On day 28, mice inhaled gradually increasing doses of methacholine (0–40 mg/mL) for 3 min. AHR information was recorded and is presented as enhanced pause (Penh) [28].

### 4.5. Bronchoalveolar Lavage Fluid

Mice were anesthetized and sacrificed with 4% isoflurane, and the trachea was intubated to flush the airway and lungs three times. Subsequently, BALF was collected to measure the concentrations of cytokines and chemokines. BALF cells were stained with Giemsa stain to enable calculation of the types and number of immune cells [29].

### 4.6. Histopathological Analysis of Lung

Lung tissues were fixed with formalin and embedded in paraffin. Lung biopsies were stained with periodic acid-Schiff (PAS) to detect goblet cell hyperplasia in the trachea. Additionally, the biopsies were stained with hematoxylin and eosin (HE) to assay eosinophil infiltration of the lungs [30].

### 4.7. Serum Analysis and Splenocyte Culture

Mice were anesthetized with isoflurane, and blood was collected from the retro-orbital plexus. Serum antibodies were detected using ELISA. Furthermore, 1 mL splenocytes (5 × 10^6^ cells/mL) were isolated and cultured in RPMI 1640 medium containing 100 μg/mL OVA for 5 days. The culture medium was collected to detect cytokine production with ELISA [13].

### 4.8. BEAS-2B Cell Culture and Gypenoside A Treatment

Human bronchial epithelial cells of line BEAS-2B (American Type Culture Collection, Manassas, VA, USA) were cultured in DMEM/F12 medium. 1 mL BEAS-2B cells (2 × 10^5^ cells/mL) were seeded into 24-well culture plates. Cells were initially treated with gypenoside A (0–10 μM) for 1 h. Next, cells were stimulated with 10 ng/mL TNF-α and 10 ng/mL IL-4 and incubated for 24 h. The culture medium was collected for ELISA assays of cytokine and chemokine production.

### 4.9. ELISA Assay

Cytokines and chemokines were detected using specific ELISA kits (R&D Systems, Minneapolis, MN, USA) according to the manufacturer’s instructions. In serum, OVA-specific antibodies, included OVA-IgG1, OVA-IgG2a, and OVA-IgE, were measured by use of a Mouse ELISA kit (BD Biosciences, San Diego, CA, USA). OVA-IgG1 and OVA-IgG2a standard curves were made using the serum of OVA-sensitized mice [31].

### 4.10. Reactive Oxygen Species Detection

BEAS-2B cells were treated with gypenoside A and incubated with TNF-α/IL-4. Subsequently, cells were added to 1 μM 6-carboxy-2-7 dichlorodihydroxyfluorescein diacetate (DCFDA) and incubated for 30 min. Reactive oxygen species (ROS) were detected using fluorescence microscopy (Olympus, Tokyo, Japan). Additionally, cells were lysed and ROS were assayed using a multi-mode microplate reader (Molecular Devices, San Jose, CA, USA).

### 4.11. Quantitative Real-Time PCR Analysis

Lung tissues were homogenized and RNA was isolated using TRI reagent (Sigma, St. Louis, MO, USA). RNA was reverse transcribed with a Reverse Transcription Kit (Thermo, Waltham, MA, USA). Quantitative real-time PCR analysis was performed with a spectrofluorometric thermocycler (Bio-Rad, San Francisco, CA, USA) and a SYBR Green PCR Master Mix kit (Thermo), as described previously [30,32]. The sequences of primer were presented in Table 1 [33,34].

### 4.12. MDA Activity and Glutathione Assay

MDA activity and glutathione were measured with a lipid peroxidation assay kit and glutathione assay kit (Sigma), respectively. Briefly, lung tissues were homogenized and incubated with the appropriate reaction solution, as previously described [29]. MDA activity and glutathione levels were measured by use of a multi-mode microplate reader (Molecular Devices).

### 4.13. Statistical Analysis

Statistical analyses were performed with ANOVAs and the Tukey–Kramer post hoc test. Data are presented as the mean ± SEM, and all results represent at least three independent experiments. *p*-values < 0.05 were considered significant.

## 5. Conclusions

In conclusion, gypenoside A significantly inhibited inflammation and oxidation in the lungs of asthmatic mice. In addition, gypenoside A inhibited Th2 cell activation, tracheal goblet cell hyperplasia, and eosinophil infiltration, and improved AHR in asthmatic mice. Thus, our findings suggest that the natural compound gypenoside A has the potential to improve asthma symptoms.

## Figures and Tables

**Figure 1 ijms-23-07699-f001:**
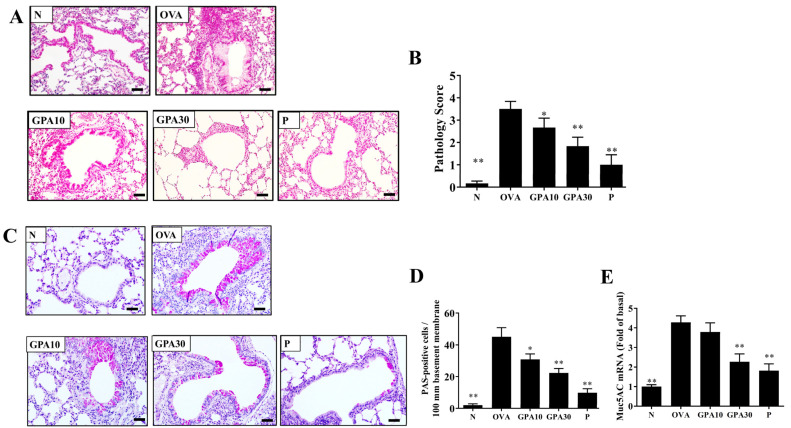
Effect of gypenoside A (GPA) on lung function. (**A**) Gypenoside A reduced eosinophil infiltration (HE stain, 200× magnification) (scale bar = 100 µm). (**B**) Pathological scores in lung tissue. (**C**) PAS-stained lung sections show goblet cell hyperplasia (200× magnification) (scale bar = 100 µm). (**D**) The number of PAS-positive cells per 100 μm of basement membrane. (**E**) Muc5AC expression levels in lung tissue. Three independent experiments were analyzed, and all data are presented as the mean ± SEM (n = 4–6 per group). * *p* < 0.05 and ** *p* < 0.01 compared with the OVA control group. Normal saline control group were named as N; Ovalbumin (OVA)-induced asthma mice were named as OVA. The 10 mg/kg and 30 mg/kg gypenoside A were named as GPA10 and GPA30, respectively. The 5 mg/kg prednisolone was named as P.

**Figure 2 ijms-23-07699-f002:**
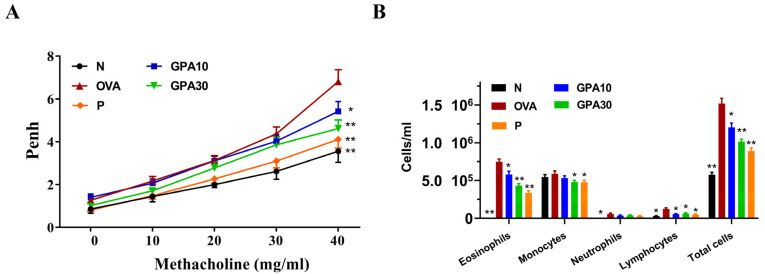
Effect of gypenoside A (GPA) on lung function and cell counts in BALF. (**A**) AHR was assessed and is shown as Penh values, and (**B**) inflammatory cells in BALF were counted. Three independent experiments were analyzed, and all data are presented as the mean ± SEM. * *p* < 0.05 and ** *p* < 0.01 compared with the OVA control group (n = 8 per group). Normal saline control group were named as N; Ovalbumin (OVA)-induced asthma mice were named as OVA. The 10 mg/kg and 30 mg/kg gypenoside A were named as GPA10 and GPA30, respectively. The 5 mg/kg prednisolone was named as P.

**Figure 3 ijms-23-07699-f003:**
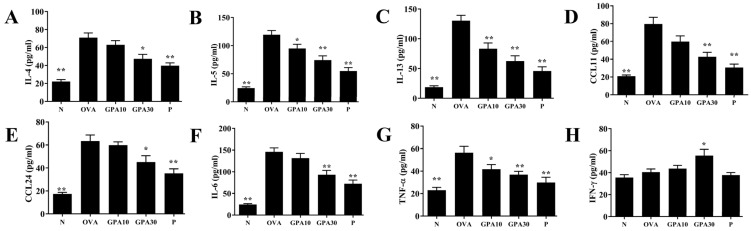
Gypenoside A (GPA) regulates BALF cytokine and chemokine levels. (**A**) IL-4, (**B**) IL-5, (**C**) IL-13, (**D**) CCL11, (**E**) CCL24, (**F**) IL-6, (**G**) TNF-α, and (**H**) IFN-γ as measured by ELISA. The data are presented as the mean ± SEM of three independent experiments (n = 8 per group). * *p* < 0.05, ** *p* < 0.01 compared with the OVA control group. Normal saline control group were named as N; Ovalbumin (OVA)-induced asthma mice were named as OVA. The 10 mg/kg and 30 mg/kg gypenoside A were named as GPA10 and GPA30, respectively. The 5 mg/kg prednisolone was named as P.

**Figure 4 ijms-23-07699-f004:**
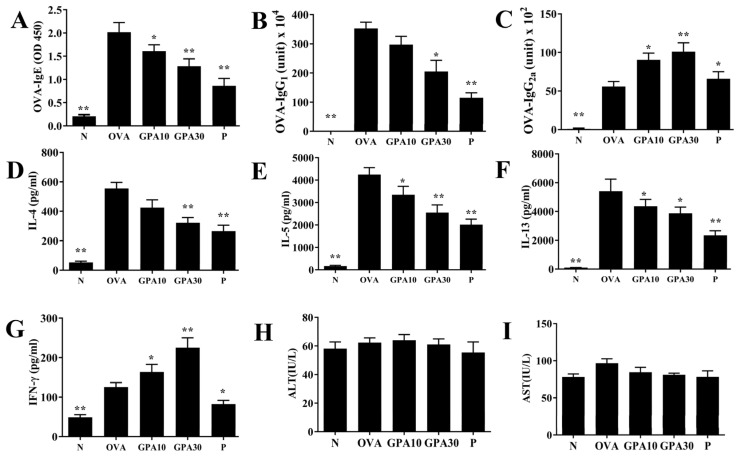
Effects of gypenoside A (GPA) on the levels of cytokines and antibodies in serum and splenocytes. Serum levels of (**A**) OVA-IgE, (**B**) OVA-IgG1, and (**C**) OVA-IgG2a in mice. Gypenoside A modulated the levels of (**D**) IL-4, (**E**) IL-5, (**F**) IL-13, and (**G**) IFN-γ produced by OVA-activated splenocytes. Effects of gypenoside A on serum biochemical value, including (**H**) ALT and (**I**) AST. The data are presented as the mean ± SEM of three independent experiments (n = 8 per group). * *p* < 0.05, ** *p* < 0.01 compared with the OVA control group. Normal saline control group were named as N; Ovalbumin (OVA)-induced asthma mice were named as OVA. The 10 mg/kg and 30 mg/kg gypenoside A were named as GPA10 and GPA30, respectively. The 5 mg/kg prednisolone was named as P.

**Figure 5 ijms-23-07699-f005:**
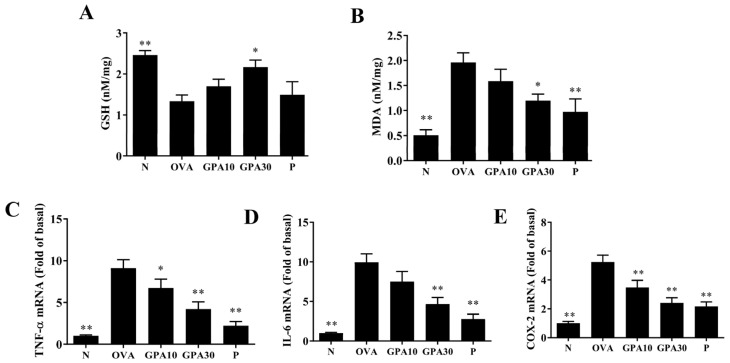
Effects of gypenoside A (GPA) on oxidative stress and inflammation in lung tissue. (**A**) GSH activity, and (**B**) MDA activity for oxidative stress. (**C**) *TNF-α*, (**D**) *IL-6*, and (**E**) *COX-2* gene expression promoting inflammation in lung tissue. The data are presented as the mean ± SEM of three independent experiments (n = 8 per group). * *p* < 0.05, ** *p* < 0.01 compared with the OVA control group. Normal saline control group were named as N; Ovalbumin (OVA)-induced asthma mice were named as OVA. The 10 mg/kg and 30 mg/kg gypenoside A were named as GPA10 and GPA30, respectively. The 5 mg/kg prednisolone was named as P.

**Figure 6 ijms-23-07699-f006:**
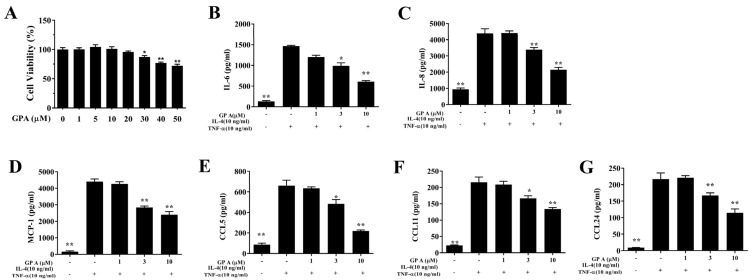
Effects of gypenoside A (GPA) on cytokine and chemokine production in BEAS-2B cells. (**A**) Cell viability with increasing concentrations of gypenoside A. ELISA showing (**B**) IL-6, (**C**) IL-8, (**D**) MCP-1, (**E**) CCL5, (**F**) CCL11, and (**G**) CCL24 levels in BEAS-2B cells. The data are presented as the mean ± SEM of three independent experiments (n = 12 per group). * *p* < 0.05, ** *p* < 0.01 compared to BEAS-2B cells stimulated with TNF-α /IL-4.

**Figure 7 ijms-23-07699-f007:**
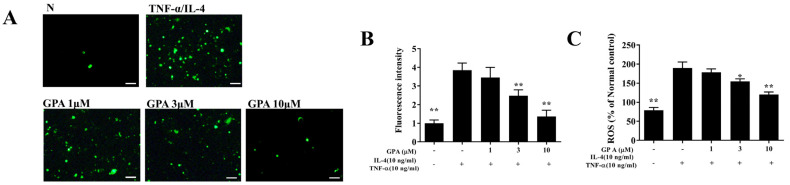
Effects of gypenoside A (GPA) on ROS production in activated BEAS-2B cells. (**A**) Fluorescence microscopy images of intracellular ROS (scale bar = 100 µm). (**B**) Fluorescence intensity of intracellular ROS. (**C**) ROS levels detected in TNF-α/IL−4-activated BEAS-2B cells with or without gypenoside A as percentages of the levels in untreated cells (N). The data are presented as the mean ± SEM of three independent experiments. * *p* < 0.05, ** *p* < 0.01 compared to BEAS-2B cells stimulated with TNF-α /IL-4.

**Table 1 ijms-23-07699-t001:** Sequences of primer pairs used for real-time PCR. Forward (F); Reverse (R).

Gene	Primer	5′–3′ Sequence
COX-2	F R	ACCAGCAGTTCCAGTATCAGA CAGGAGGATGGAGTTGTTGTAG
IL-6	F	AGGACCAAGACCATCCAATTCA
	R	GCTTAGGCATAACGCACTAGG
Muc5AC	F R	AATGCTGGTGCCTGTGTCTCAGAGGGA CCTCCTATGCCATCTGTTGTG
TNF-α	F	GCACCACCATCAAGGACTC
	R	AGGCAACCTGACCACTCTC
β-actin	F	AAGACCTCTATGCCAACACAGT
	R	AGCCAGAGCAGTAATCTCCTTC

## Data Availability

The data presented in this study is available on request from the corresponding author.

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
