# Peer review of "Gypenoside A from Gynostemma pentaphyllum Attenuates Airway Inflammation and Th2 Cell Activities in a Murine Asthma Model"

_ijms, 2022, doi:10.3390/ijms23147699_

Round 1
Reviewer 1 Report
This manuscript contains extensive amount of in vivo and in vitro data which support the inhibitory effect of gypenoside A on the development of airway inflammation in a murine model of asthma bronchiale. The Authors also provide mechanistic data to explain the described observations. The data presented here is clinically relevant and the manuscript is well written. However, a few points should be clarified before the final publication of the manuscript.
Specific suggestions:
1. Due to the current preference in scientific literature of a non-stigmatizing language to describe diseases, adjectives should be avoided. Therefore, instead of “asthmatic patients”, “patients with asthma” should be written.
2. L68: “which” instead of “what”.
3. L70: “gypenoside A, a triterpenoid isolated from G. pentaphyllum” a reference is missing here.
4. L82: BALF should be defined here not in L253.
5. I suggest combining Figures 1 and 2.
6. L108: “Secretion” instead of “expression”.
7. L129: MDA should be defined here not in L291.
8. L172-4173: The sentence should be rephrased for clarification.
9. 4.3. The dose of prednisolone and metacholine should be included.
10. 4.7. How many splenocytes, in what volume, and in which type of medium were incubated with OVA?
11. 4.8. How many cells and in what volume were treated?
12. 4.10. The concentration of DCFDA should be included.
13. 4.11. What was the source of the PCR primers/assays?
Author Response
This manuscript contains extensive amount of in vivo and in vitro data which support the inhibitory effect of gypenoside A on the development of airway inflammation in a murine model of asthma bronchiale. The Authors also provide mechanistic data to explain the described observations. The data presented here is clinically relevant and the manuscript is well written. However, a few points should be clarified before the final publication of the manuscript.
Specific suggestions:
- Due to the current preference in scientific literature of a non-stigmatizing language to describe diseases, adjectives should be avoided. Therefore, instead of “asthmatic patients”, “patients with asthma” should be written.
Responses:
Thank for reviewer’s suggestion. In this manuscript, Patients with asthma instead of asthmatic patients.
- L68: “which” instead of “what”.
Responses:
We have made corrections in this manuscript. (Line 67 However, it is not clear which active compounds of G. pentaphyllum might improve airway inflammation or oxidative stress in asthmatic mice)
- L70: “gypenoside A, a triterpenoid isolated from G. pentaphyllum” a reference is missing here.
Responses:
We added a reference in this manuscript.(Line 70)
- L82: BALF should be defined here not in L253.
Responses:
Thank for reviewer’s suggestion.
bronchoalveolar lavage fluid (BALF) were collected and described on line 85-86.
- I suggest combining Figures 1 and 2.
Responses:
Thank for reviewer’s suggestion. We combine Figures 1 and 2.
- L108: “Secretion” instead of “expression”.
Responses:
Thank for reviewer’s suggestion. We modified as “Gypenoside A regulates chemokine and cytokine secretion in BALF” (Line 111)
- L129: MDA should be defined here not in L291.
Responses:
Thank for reviewer’s suggestion. We defined malondialdehyde (MDA) on line 137-138.
- L172-173: The sentence should be rephrased for clarification.
Responses:
The sentence modified as “IL-13 knockout mice induced asthma, and the mice did not increase AHR and goblet cell hyperplasia”. (Line 183-184)
- 4.3. The dose of prednisolone and metacholine should be included.
Responses:
5 mg/kg prednisolone was described on line 246-247
methacholine (0–40 mg/mL) was described on line 259.
- 4.7. How many splenocytes, in what volume, and in which type of medium were incubated with OVA?
Responses:
We add those describe in line 273-275.
- 4.8. How many cells and in what volume were treated?
Responses:
We add those describe in line 279-280.
- 4.10. The concentration of DCFDA should be included.
Responses:
We add those describe in line 292.
- 4.11. What was the source of the PCR primers/assays?
Responses:
We add Table 1 about PCR primers.

Reviewer 2 Report
This study reports that gypenoside A, derived from G. pentaphyllum, reduces airway hyperresponsiveness and various inflammatory and immunomodulatory (Th2-type) cytokines in a mouse asthma model. The data appear to convincingly show that this natural product may have anti-allergic/inflammatory properties and, interestingly, also reduces allergen-specific IgE in this model.
My main issue with this work, however, is that it is rather similar to those previously published by the authors in Food Chem Toxicol. 2010 Oct;48(10):2592-8 and in Am J Chin Med. 2008;36(3):579-92. While the authors have clearly mentioned these earlier studies, other than now testing the effects of gypenoside A rather than an extract from G. pentaphyllum, the findings are relatively incremental and do not offer important new insights. For example, there are no comparisons to show whether gypenoside A is more effective than other gypenosides or how it compares to G. pentaphyllum extract. Furthermore, the authors have not carried out any investigations to shed light on the potential mechanism of action (i.e. which intracellular signals are involved/targeted?). At the end of the discussion the authors state that “gypenoside A is an effective antioxidant” but antioxidant mechanisms alone are very unlikely to explain the inhibition of cytokine, chemokine and allergen-specific immunoglobulin levels/in vitro production. Moreover, the potential anti-allergic/inflammatory effects of gypenoside A are not superior to prednisolone. What would be the benefits of gypenoside A therapy compared to prednisolone? What are the potential side-effects/adverse drug reactions? The authors should mention any toxicological properties and also investigate whether gypenoside A affects cell viability and apoptosis in lung tissues and BEAS-2B cells.
Other issues
Abstract, line 25, and elsewhere: signs not symptoms (these observations are in mice)
Figs 1-6: There is no indication what “N” or “P” or the concentration units for gypenoside A mean. These should be stated in the results section and in the figure legend (at least on first use) and not just hidden in the methods section.
Discussion, line 172. Rewrite the sentence “House dust mite–induced IL-13-deficient mice, the mice did not be induced to promote AHR [17].”
Author Response
This study reports that gypenoside A, derived from G. pentaphyllum, reduces airway hyperresponsiveness and various inflammatory and immunomodulatory (Th2-type) cytokines in a mouse asthma model. The data appear to convincingly show that this natural product may have anti-allergic/inflammatory properties and, interestingly, also reduces allergen-specific IgE in this model.
(1)My main issue with this work, however, is that it is rather similar to those previously published by the authors in Food Chem Toxicol. 2010 Oct;48(10):2592-8 and in Am J Chin Med. 2008;36(3):579-92. While the authors have clearly mentioned these earlier studies, other than now testing the effects of gypenoside A rather than an extract from G. pentaphyllum, the findings are relatively incremental and do not offer important new insights. For example, there are no comparisons to show whether gypenoside A is more effective than other gypenosides or how it compares to G. pentaphyllum extract. (2)Furthermore, the authors have not carried out any investigations to shed light on the potential mechanism of action (i.e. which intracellular signals are involved/targeted?). (3) At the end of the discussion the authors state that “gypenoside A is an effective antioxidant” but antioxidant mechanisms alone are very unlikely to explain the inhibition of cytokine, chemokine and allergen-specific immunoglobulin levels/in vitro production. (4)Moreover, the potential anti-allergic/inflammatory effects of gypenoside A are not superior to prednisolone. What would be the benefits of gypenoside A therapy compared to prednisolone? (5)What are the potential side-effects/adverse drug reactions? The authors should mention any toxicological properties and (6)also investigate whether gypenoside A affects cell viability and apoptosis in lung tissues and BEAS-2B cells.
(1) Responses: Our previous report demonstrated that the oral administration of short-term high dose Gynostemma pentaphyllum extract (5 g/kg per day for 7 days) decreased allergic reactions in ovalbumin (OVA)-sensitized mice. We found that G. pentaphyllum extract significantly decreased airway hyperresponsiveness, and suppressed eosinophil infiltration in the lung tissue of asthmatic mice. The extract also reduced the levels of Th2 cytokines and chemokines in bronchoalveolar lavage fluid and splenocytes, and suppressed OVA–IgE production in serum. Our laboratory does not have the ability to isolate the pure compounds of G. pentaphyllum or other herbs. Therefore, our previous experiments can only use G. pentaphyllum extract for asthma experiments. In 2019, we found that there were already many novel purified compounds of G. pentaphyllum sold by biotech company. Therefore, we purchased several pure compounds of G. pentaphyllum to test the inflammatory response in tracheal epithelial cells. Those pure compounds include Gypenoside A, Gypenoside L, Gypenoside LI, Gypenoside XIII, Gypenoside XLIX, Gypenoside XLVI, and Gypenoside XVII. We found that Gypenoside A most inhibited inflammatory cytokine secretion in inflamed BEAS-2B cells. Therefore, in this experiment, we used Gypenoside A to investigate asthma experiments in asthmatic mice. In this experiment, we not only investigated AHR and eosinophil infiltration and the secretion of Th2 cytokines in spleen cellsand BALF, we also investigated that Gypenoside A reduced tracheal goblet cell hyperplasia to inhibit mucus hypersecretion. In addition, in this experiment we also investigated the inflammatory response and oxidative stress of airway and tracheal epithelial cells. In 2008 and 2010, we used 5g/kg of G. pentaphyllum crude extract in our asthma experiment. However, in this experiment, Gypenoside A 10 mg/kg and 30 mg/kg were used for asthma mice experiments to reduce unnecessary substance intake. We thought that Gypenoside A treatment significantly reduced AHR, eosinophil infiltration, goblet cell hyperplasia, and airway inflammation in the lungs of asthmatic mice. Gypenoside A inhibited oxidative stress in the lungs of asthmatic mice. Gypenoside A also suppressed levels of Th2 cytokines and chemokines in BALF and lung tissue. We confirmed that Gypenoside A decreased pro-inflammatory cytokine, chemokine, and eotaxin expression in inflammatory BEAS-2B cells.
(2) Furthermore, the authors have not carried out any investigations to shed light on the potential mechanism of action (i.e. which intracellular signals are involved/targeted?).
(2) Responses: This study investigated the effect of Gypenoside A on modulating immune responses and reducing airway inflammation in asthmatic mice. Thank for reviewer’s suggestion. Our laboratory is still investigating whether Gypenoside A improves the molecular mechanism of inflammasome and autophagy in asthmatic mice. In addition, we continued to evaluate the molecular mechanism of Gypenoside A regulating innate lymphoid cell II (ILC II) and inflammasome in obese asthmatic mice.
(3) Gypenoside A is an effective antioxidant” but antioxidant mechanisms alone are very unlikely to explain the inhibition of cytokine, chemokine and allergen-specific immunoglobulin levels/in vitro production.
(3) Responses: Previous studies pointed out that ROS could increase AHR, tracheal smooth muscle contraction, and mucus secretion. The antioxidant enzymes, SOD, and glutathione can remove ROS in the lungs, which reduces lung cell damage and fibrosis in patients with asthma. Oxidative damage also causes lipid peroxidation and high levels of malondialdehyde in the lungs of patients with asthma. We found that Gypenoside A promoted pulmonary glutathione expression and reduced malondialdehyde expression in asthmatic mice. Thus, our findings suggested that Gypenoside A protected against oxidative stress and reduced oxidative stress-related damage to improve lung function.
(4) Moreover, the potential anti-allergic/inflammatory effects of gypenoside A are not superior to prednisolone. What would be the benefits of gypenoside A therapy compared to prednisolone?
(4) Responses: Clinically, inhaled steroids and oral steroids are commonly used to treat or prevent asthma, but some patients have side effects on steroids. The use of steroid drugs is limited and invalid for treating neutrophilic asthma or severe asthma patients. Therefore, the development of new effective drugs for treating asthma is expected by many researchers and clinicians. Our experiments show that asthmatic mice treated with prednisolone should have better reductions in AHR, eosinophilic infiltration, goblet cell hyperplasia, and airway inflammation than Gypenoside A-treated asthmatic mice. However, steroids are an immunosuppressant that inhibits the activation of immune cells. Therefore, steroid-treated asthmatic mice would reduce the levels of Th1 cell-associated cytokine IFN-r in BALF and spleen cells. Steroid also reduced OVA-IgG2a, Th1 cell activation-related antibody, in serum. However, Gypenoside A can increase IFN-r secretion in BALF and spleen cells, and increase the levels of OVA-IgG2a in serum. Apparently, Gypenoside A is not an immunosuppressant. Gypenoside A has the effect of regulating immune cells to improve asthma.
(5) What are the potential side-effects/adverse drug reactions? The authors should mention any toxicological properties
(5) Responses: In this experiment, we detected serum ALT AST in asthmatic mice. Gypenoside A could decrease ALT and AST levels in serum of asthmatic mice (Experimental data not shown). Hence, Gypenoside A is not hepatotoxic in asthmatic mice.
(6) investigate whether gypenoside A affects cell viability and apoptosis in lung tissues and BEAS-2B cells.
(6) Responses: We did not analyze cellular activity in lung tissue. However, BEAS-2B cells were evaluated for cell viability and apoptosis by CCK8 assay and DAPI staining, respectively. Gypenoside A did not demonstrate significant cytotoxic effects at a concentration ≤20 μM, and subsequent experiments used Gypenoside A at 0–10 μM. 10 uM Gypenoside A also did not induce DNA condensation in BEAS-2B cells (Experimental data not shown). Therefore, the concentrations of Gypenoside A did not affect cell survival and apoptosis.
Other issues
Abstract, line 25, and elsewhere: signs not symptoms (these observations are in mice)
Responses:
Thank for reviewer’s suggestion. We have modified it on line 25, and elsewhere.
Figs 1-6: There is no indication what “N” or “P” or the concentration units for gypenoside A mean. These should be stated in the results section and in the figure legend (at least on first use) and not just hidden in the methods section.
Responses:
Thank for reviewer’s suggestion. We add some describe in Figure Legends (Fig 1-6)
Normal saline control group were named as N; Ovalbumin (OVA)-induced asthma mice were named as OVA. 10 mg/kg, 30 mg/kg gypenoside A were named as GPA10 and GAP30, respectively. 5 mg/kg prednisolone was named as P.
Discussion, line 172. Rewrite the sentence “House dust mite–induced IL-13-deficient mice, the mice did not be induced to promote AHR [17].”
Responses:
We modified as “ IL-13 knockout mice induced asthma, and the mice did not increase AHR and goblet cell hyperplasia”.(line 183-184)

Round 2
Reviewer 2 Report
The manuscript has improved but the authors have not mentioned the effects of gypenoside A on cell viability/apoptosis (only in their reply but the manuscript was not revised for this), nor have they addressed potential toxicological issues. I'm fully aware of the role of ROS in asthma but my point is that reduction of ROS may not be a central mechanism of action for this agent to explain the reduction of pro-allergic cytokines. Antioxidants per se are unlikley to be sufficient (otherwise ascorbic acid, for example, would bve an effective therapy).
Author Response
The manuscript has improved but the authors have not mentioned the effects of gypenoside A on cell viability/apoptosis (only in their reply but the manuscript was not revised for this), nor have they addressed potential toxicological issues.
Responses:
Thank for reviewer’s suggestion. We add the result of cell viability in Figure 6A, and add ALT and AST levels of serum in Figure 4H and 4I.
I'm fully aware of the role of ROS in asthma but my point is that reduction of ROS may not be a central mechanism of action for this agent to explain the reduction of pro-allergic cytokines. Antioxidants per se are unlikley to be sufficient (otherwise ascorbic acid, for example, would bve an effective therapy).
Responses: Someone evaluate the antioxidant capacity of pure compounds using the DPPH method to detect the scavenging capacity of free radicals. The benefit of this method is to detect free radical scavenging activities in a cell-free state. Here, in our experiments, BEAS-2B human bronchial epithelial cells were treated with various concentrations of Gypenoside A and then stimulated with TNF-α/ IL-4. In addition, ovalbumin-sensitized mice were treated with Gypenoside A to detect inflammatory mediators and oxidative stress expression. Hence, we used DCFDA assay, not DPPH, to evaluate reactive oxygen species in BEAS-2B cells. Furthermore, we detect MDA activity and glutathione assay in lung of asthmatic mice. Asthma attacks can induce oxidative stress in the lung and attenuate lung function. GSH is the most abundant intracellular thiol-based antioxidant, and plays an important role in the cellular defense cascade against oxidative injury. Lipid peroxidation is a danger signal of cell damage and would produce abundant MDA as a marker of oxidative stress in cells and tissues. Our experiments found that Gypenoside A could promote GSH expression and reduce MDA levels in lung of asthmatic mice. We thought that the antioxidant experiments of cell and animal experiments can show the antioxidant capacity of Gypenoside A. Hence, we did not detect the scavenging capacity of free radicals using DPPH method. Our findings suggested that Gypenoside A protected against oxidative stress and reduced oxidative stress-related damage to improve lung function in asthmatic mice.
